MALDI-TOF MS as a method for rapid identification of Phytophthora de Bary, 1876

Božik Matěj bozik@af.czu.cz 1
Mrázková Marcela 2
Novotná Karolína 1
Hrabětová Markéta 2
Maršik Petr 1
Klouček Pavel 1
Černý Karel Karel.Cerny@vukoz.cz 2
1 Czech University of Life Sciences, Department of Food Science , Prague , Czech Republic
2 The Silva Tarouca Research Institute for Landscape and Ornamental Gardening , Pruhonice , Czech Republic
Portillo Maria del Carmen
Electronic publication date: 2021 Jul 19
Publication date: 2021
Volume: 9
Electronic Location ID: e11662
Received 2021 Mar 25; Accepted 2021 Jun 1
Copyright: ©2021 Božik et al.
Copyright year: 2021
Copyright holder: Božik et al.
License: This is an open access article distributed under the terms of the Creative Commons Attribution License, which permits unrestricted use, distribution, reproduction and adaptation in any medium and for any purpose provided that it is properly attributed. For attribution, the original author(s), title, publication source (PeerJ) and either DOI or URL of the article must be cited.
License URL: https://creativecommons.org/licenses/by/4.0/

Keywords: Phytophthora, Maldi, Biotyping, Species determination, ITS, COI, Library, Database, Mass spectra, Protein

Funding: The internal grant agency of Czech University of Life Sciences Prague (CIGA) Project no. 20182019 The European Regional Development Fund-Project Centre for the investigation of synthesis and transformation of nutritional substances in the food chain in interaction with potentially harmful substances of anthropogenic origin: comprehensive assessment of soil contamination risks for the quality of agricultural products No. CZ.02.1.01/0.0/0.0/16_019/0000845 The Research Infrastructure METROFOOD-CZ supported by the Ministry of Education Youth and Sports of the Czech Republic under Project No. LM2018100 The Ministry of the Environment of the Czech Republic under Project No. IP–00027073–VUKOZ This work was supported by the internal grant agency of Czech University of Life Sciences Prague (CIGA) Project no. 20182019 and the European Regional Development Fund-Project Centre for the investigation of synthesis and transformation of nutritional substances in the food chain in interaction with potentially harmful substances of anthropogenic origin: comprehensive assessment of soil contamination risks for the quality of agricultural products (No. CZ.02.1.01/0.0/0.0/16_019/0000845). The authors received support and assistance from the Research Infrastructure METROFOOD-CZ supported by the Ministry of Education, Youth and Sports of the Czech Republic under Project No. LM2018100 and the Ministry of the Environment of the Czech Republic under Project No. IP–00027073–VUKOZ. The funders had no role in study design, data collection and analysis, decision to publish, or preparation of the manuscript.

==============================
The number of described species of the oomycete genus Phytophthora is growing rapidly, highlighting the need for low-cost, rapid tools for species identification. Here, a collection of 24 Phytophthora species (42 samples) from natural as well as anthropogenic habitats were genetically identified using the internal transcribed spacer (ITS) and cytochrome c oxidase subunit I (COI) regions. Because genetic identification is time consuming, we have created a complementary method based on by matrix-assisted laser desorption ionization–time of flight mass spectrometry (MALDI-TOF MS). Both methods were compared and hypothesis that the MALDI-TOF MS method can be a fast and reliable method for the identification of oomycetes was confirmed. Over 3500 mass spectra were acquired, manually reviewed for quality control, and consolidated into a single reference library using the Bruker MALDI Biotyper platform. Finally, a database containing 144 main spectra (MSPs) was created and published in repository. The method presented in this study will facilitate the use of MALDI-TOF MS as a complement to existing approaches for fast, reliable identification of Phytophthora isolates.

Introduction

The oomycete genus Phytophthora was first established by de Bary in 1876 with the type species P. infestans, which causes potato blight and was the culprit in a series of famines in Europe (De Bary, 1876). Members of this genus are among the most cited pathogens worldwide and are associated with annual economic losses reaching billions of dollars (Erwin & Ribeiro, 1996). Plant pathogens belonging to Phytophthora impact agriculture, horticulture, forestry, and natural ecosystems (Hansen, Reeser & Sutton, 2012), and epidemics of the most destructive species, including P. alni, P. austrocedrae, P. cinnamomi, P. ramorum, and P. kernoviae, have occurred in various parts of the world (Greslebin, Hansen & Sutton, 2007). The number of described Phytophthora species grew from 58 in 1996 (Erwin & Ribeiro, 1996) to 313 in 2018 (H Ho, 2018), followed by a dramatic spike in recent years to an estimated total of 600 (Brasier, 2009). New species have been split from old species as species complexes were resolved but are also arising from the recent hybridization of existing species such as Phytophthora alni subsp. alni (Husson et al., 2015; Mizeriene et al., 2020). Some of these new species have been described as causal agents of new diseases, while others have been discovered through the exploration of new habitats.

This explosion in diversity is also the result of applying new molecular tools to taxonomy (Hansen, Reeser & Sutton, 2012). Traditionally, species descriptions were based on morphological characteristics, but phenotypic species identification is time-consuming and requires specialized scientific knowledge (Ruiz Gómez et al., 2019). Studies of soil fungal and oomycetous communities have benefitted from modern techniques like next-generation sequencing (NGS) (Hardham & Blackman, 2018), pyrosequencing and second-generation techniques such as metabarcoding (Prigigallo et al., 2016) based on Illumina de novo sequencing. Although morphological characteristics still serve as the basis for the preliminary orientation and identification of Phytophthora species, various genetic methods are now routinely used when overlapping morphological features and intraspecific variability present a challenge for Phytophthora species identification (Martin et al., 2012).

In these techniques, species identification typically depends on sequencing of the internal transcribed spacer (ITS) region of ribosomal DNA (rDNA) (Cooke et al., 2000; Grünwald et al., 2011; Sandoval-Sierra, Martín & Diéguez-Uribeondo, 2014; Jung et al., 2019). Online databases such as GenBank (http://www.ncbi.nlm.nih.gov) and EMBL (http://www.ebi.ac.uk) collect sequence data and allow species identification by BLAST based primarily on the coefficient of similarity of DNA and RNA sequences (Ristaino, 2012; Ho, 2018). For the genus Phytophthora, the ”Phytophthora Database” (http://www.Phytophthoradb.org) provides information on morphological features, geographic distribution, and relevant references (Park et al., 2008). The advantage of the ITS region is that its sequence can readily be obtained, and the ITS sequences of many Phytophthora species are currently available in GenBank. However, a disadvantage of ITS sequences is that there are minimal or no differences between closely related species, such as P. rubi and P. fragariae (Martin et al., 2012), and intraspecific variation can blur the boundaries between some species. Additional genetic regions that are commonly used for species identification include the mitochondrial cytochrome c oxidase subunit genes (COX1 and COX2) (Martin & Tooley, 2003; Kroon et al., 2004; Robideau et al., 2011), but the need to sequence additional regions further increases the cost, which is a limiting factor for identification based on DNA sequencing (Del Castillo-Múnera et al., 2013).

A potential alternative to sequencing is matrix-assisted laser desorption ionization–time of flight mass spectrometry (MALDI-TOF MS). MALDI-TOF MS is a fast and direct form of analysis that requires no chromatographic separation (Galeano Garcia et al., 2018). Recent studies have shown that MALDI biotyping is a fast and reliable method of species identification for mushrooms (Sugawara et al., 2016; El Karkouri et al., 2019), yeasts, molds (Rizzato et al., 2015; Drissner & Freimoser, 2017), mycorrhizal fungi (Crossay et al., 2017) and the pathogenic oomycete Pythium insidiosum (Krajaejun et al., 2018).

In this study, we focused on 43 selected strains belonging to 26 Phytophthora species and one species belonging to the genus Pythium as an outgroup. The strain collection was created from isolates acquired from natural as well as anthropogenic (nurseries, gardening centers, fruit orchards, etc.) habitats. The isolates were subjected to genetic identification using the cytochrome c oxidase subunit I (COI) and ITS regions, and the validated strains were grown in liquid medium and subsequently analyzed by MALDI-TOF MS (Fig. S1). Finally, a library of 144 main spectra (MSPs) was created and validated.

Materials & Methods

Pathogen isolation, cultivation, genetic identification, and preservation

Oomycetous pathogens were isolated from damaged plants of different ornamental, forest and fruit woody plant species and from water in 2008–2017. Woody plants of many species from different stands (nurseries, orchards, gardening centers and ornamental gardens, forest and riparian stands) with disease symptoms such as yellowing or wilting foliage, dieback, root and collar rot, or rot of feeding roots were identified and examined for the presence of damaged plant tissues. Tissues exhibiting characteristic lesions were aseptically sampled and transported to the laboratory. The samples were rinsed under tap water, and transition zones between necrotized and healthy plant tissues were identified. Segments (3 ×3 mm) from the edges of active lesions were excised, surface disinfected by immersion in 95% ethanol for 10 s, rinsed in deionized sterile water and dried with sterile filter paper. Then, the segments were placed on selective PARPNH agar medium (V8 juice 200 mL, ampicillin 200 ppm, rifampicin 10 ppm, quintozene 25 ppm, nystatin 50 ppm, hymexazol 50 ppm, agar 15 g, CaCO3 3 g, and deionized water 800 mL) in 9-cm Petri dishes and incubated at 20 °C in the dark. After two to five days, segments exhibiting characteristic oomycetous hyphae were transferred onto plates containing V8A medium (V8 juice 200 mL, agar 15 g, CaCO3 3 g, deionized water 800 mL). Pure isolates were transferred to oatmeal agar (HiMedia Ltd, India) slants in tubes and stored in refrigeration boxes at 12 °C in the Czech Collection of Phytopathogenic Oomycetes (CCPO https://www.vukoz.cz/index.php/en/collections/collection-of-phytopathogenic-oomycetes-of-rilog).

Samples of soil substrate from plants with damaged feeding roots were collected from three sites at depths of 10–20 cm in the root zone of the affected plants. A mixed sample of approximately 1,000–2,000 cm3 was then created from a single plant and transported to the laboratory. Soil samples were processed via a baiting method. Approximately 150 cm3 of soil substrate containing damaged roots was placed in a deep plastic bowl, and deionized water was added to cover the sample by 2–3 cm. The bait (young, healthy, rhododendron leaves of the susceptible Rhododendron yakushimanum cv. Silberwolke) was rinsed under tap water, placed on the water surface and incubated at room temperature under natural light in a standard day/night regime. When characteristic lesions appeared on the baits, the segments with necrotized tissues were excised and processed as described above. Floating oomycetous zoospores were trapped from water in situ using healthy leaves of R. yakushimanum cv. Silberwolke placed in a sterilized bag made from loosely woven fabric. Four bags containing leaf bait were allowed to float just under the surface of the investigated watercourses, and after one week of incubation, the samples were collected, transported to the laboratory and processed as leaf baits as described above.

Total genomic DNA of the isolates was extracted from pure cultures using a DNeasy UltraClean Microbial Kit (QIAGEN, DE) according to the manufacturer’s instructions. For molecular identification, the nuclear rDNA ITS region and/or the COI region were amplified by polymerase chain reaction (PCR). The primer pairs used to amplify the ITS region were ITS1/ITS4 or ITS4/ITS5 (White et al., 1990), and the COI region was amplified with OomCoxILevup/Fm85mod primers (Robideau et al., 2011). PCR was performed in a Mastercycler Nexus Gradient GSX 1 thermal cycler (Eppendorf, DE), and products were visualized via agarose gel electrophoresis in 1% TBE buffer using a 100-bp DNA ladder (New England Biolabs, USA) as a size marker. PCR products were purified and sequenced in both directions by Macrogen Europe B.V. (NL) using the same primers used for PCR amplification. The obtained sequences were edited and aligned in BIOEDIT (Hall, 1999) and compared with sequences in GenBank using a BLAST similarity search (http://www.ncbi.nlm.nih.gov). All sequences obtained in this study were deposited in the NCBI GenBank database (Table S1).

MALDI-TOF MS biotyping

After genetic identification, cultures of 43 selected strains belonging to 26 taxa of Phytophthora and one species belonging to the genus Pythium as an outgroup were incubated in 0.5 L of Tryptone Soya Broth (TSB, Sigma Aldrich, USA) with shaking in the dark for 3–7 days (according to the intensity of growth of each species) to develop sufficient mycelial biomass. Samples from the liquid nutrient medium were transferred to 70% ethanol and stored in Eppendorf tubes at −20 °C until analysis. An overview of the species, including their collection numbers, is given in Table S1.

For protein extraction and subsequent MALDI-TOF MS, mycelium was harvested from the liquid nutrient medium samples by centrifugation in a 1.5-mL Eppendorf tube and divided into small, approximately equal-sized pieces of 5–10 mg using micro scissors. Distilled water (300 µL) was added to each piece of mycelium and mixed thoroughly. Then, 900 µL of ethanol was added, followed by thorough mixing. The sample was centrifuged at 20000 × g for 2 minutes (Rotanta 450 R, Hettich, DE), the supernatant was decanted, and the sample was briefly centrifuged again. The rest of the supernatant was carefully removed from the pellet by pipetting, and the sample was subsequently allowed to dry at room temperature. Next, 10–20 µL of 70% formic acid (Sigma-Aldrich, DE) was added to fully immerse the dry pellet, followed by thorough mixing and incubation for 3 min. Finally, an equal volume of ultrapure acetonitrile (Fluka, DE) was added, followed by thorough mixing and centrifugation at 20000 × g for 2 minutes. One microliter of the supernatant was immediately applied to a spot of an MTP 384 ground steel BC target plate for MALDI-TOF MS (Bruker, DE) and allowed to dry at room temperature. Immediately after drying, 1 µL of HCCA (α-cyano-4-hydroxycinnamic acid, Bruker, DE) matrix solution (acetonitrile 50%, water 47.5% and trifluoracetic acid 2.5%; 10 mg/mL HCCA) was applied to each spot and allowed to dry again at room temperature. Each sample was applied to 8 spots around the calibration spot. For calibration, 0.5 L of Bruker Bacterial Test Standard (BTS, Bruker, DE) was applied to the center spot, followed by the application of matrix solution as described for the supernatant. Each spot was measured three times by MALDI-TOF MS Autoflex Speed (Bruker, DE) and flexControl 3.4 (Build 135) with a standard MALDI Biotyper method (MBT_FC.par). Ion source 1 was set to 19.38 kV, ion source 2 was set to 18.18 kV, and detection was set from 2 to 20 kDa. All experiments were performed in independent biological triplicates; three spectra were measured for each sample, and each spectrum was collected as 2000 shots in 200 steps.

Data processing

For each measurement, the spectra were manually inspected by flexAnalysis 3.4 Compass 1.4 (Bruker Daltonics, DE), and then MSPs were processed by the standard MALDI Biotyper MSP creation method using MALDI Biotyper Version 3.1 (Build 66). MALDI Biotyper was used to compare the results based on the similarity of the spectra in the range of 2–20 kDa. For evaluation, the standard method of identification of MSPs was chosen; the frequency threshold for spectrum adjustment was 50, the frequency threshold for calculating the score was 5, the maximum primary spectrum mass error was 2000, the mass tolerance for the modified spectrum was 350, and the accepted mass tolerance of the peak was 600 par.

The acquired spectrum of the sample was transformed into a peak pattern utilizing dedicated spectral analysis tools. The peak pattern was then compared to reference peak lists of organisms in the reference library, and a log(score) value between 0.00 and 3.00 was calculated. In brief, identification is based on average peak intensity, peak position and peak occurrence frequency, and each factor is assigned a value between 0 and 1. The three values are then summed, and the result is normalized to 1000. The decimal logarithm of the result is taken as the calculated score, such that the maximum achievable score is 3 (= log 1000). The values are then classified. The calculated score determines the highest match at the top of the list. The higher the log(score) value, the higher the degree of similarity to a given organism in the reference library. Values ≥ 2 indicate high-confidence identifications. Log (score) values between 1.70 and 1.99 need to be confirmed by additional methods. Values below 1.70 indicate that no organism identification is possible (MALDI Biotyper 3.0 User Manual Revision 2, 2011). Data correlations were visualized as score-oriented dendrograms generated from spectra or MSPs in MALDI Biotyper Version 3.1 (Build 66) using the centroid linkage method.

Results

MSP Library

Over 3500 mass spectra were processed and manually reviewed for quality control to obtain a total of 144 MSPs from 43 different samples; these MSPs were added to a custom MSP library ( http://doi.org/10.5281/zenodo.4271724). The produced MSPs were compared with the available MBT Library BDAL 9.0 (Bruker Daltonics, DE), which includes 8,468 MSPs, and the MBT Filamentous Fungi Library V1.0 (365 MSPs). After validation and verification of the method, the samples were compared with respect to species and strain classification.

Spectra of different Phytophthora strains were searched against the reference Bruker Library, which contains 2,969 species and 8,468 MSPs of various microorganisms, mainly bacteria and fungi, but no Phytophthora strains. None of the sample mass spectra matched the reference organisms, as the obtained identification scores were less than 1.70. The MSPs of the Phytophthora strains were then added to the Bruker MALDI Biotyper Library.

Library validation and verification

The same extraction method using ethanol and formic acid was used for validation. Intraday variability was determined based on successful measurement of 8 spots of the same sample in three replicates, and interday variability was based on the similarity of the spectra of samples measured on different days. Each sample was measured at least 3 times. The procedure and conditions were always identical. Figure 1 shows a comparison of the spectra of a sample of Phytophthora lacustris with collection number 295.09. The log scores of identical samples measured on different days are very similar (score > 2.6).

Figure 1 Comparison of the interday variability of spectra of P. lacustris 295.09.

Blank analysis was performed to further validate the method. One of the 43 samples was randomly selected and measured in 3 spots by MALDI Biotyper Realtime Classification (Bruker, DE). The sample was correctly identified as P. lacustris in accordance with the previous DNA classification. The spectra of the blank sample matched the MSP in our database as shown in Table 1.

Table 1 Evaluation of rapid blank sample identification.

Repletion	ID	Best match	Score	Second match	Score	
1	Blank	Phytophthora lacustris 398.10/2	2.686	Phytophthora lacustris 398.10/1	2.624	
2	Blank	Phytophthora lacustris 398.10/2	2.725	Phytophthora lacustris 398.10/1	2.688	
3	Blank	Phytophthora lacustris 398.10/2	2.718	Phytophthora lacustris 398.10/1	2.697	

Table 2 Numbers of correctly identified species and strains with different score values.

 	Species/genus identification	Strain identification	
Species	No. of strains	Biological replicates	Score >2a	Score 1.7b	Score >2a	Score 1.7b	
Phytophthora alni subsp. alni	1	3	3	3	3	3	
Phytophthora alni subsp. uniformis	2	6	6	6	6	6	
Phytophthora cactorum	2	6	6	6	6	6	
Phytophthora cambivora	2	6	6	6	6	6	
Phytophthora cf. kelmania	1	3	3	3	3	3	
Phytophthora cinnamomi	2	6	6	6	6	6	
Phytophthora citrophthora	1	3	3	3	3	3	
Phytophthora cryptogea	2	6	6	6	6	6	
Phytophthora gonapodyides	2	6	5	6	5	6	
Phytophthora gregata	2	6	6	6	6	6	
Phytophthora hedraiandra	2	6	6	6	2	5	
Phytophthora chlamydospora× amnicola	2	6	6	6	0	6	
Phytophthora lacustris	4	12	12	12	8	10	
Phytophthora megasperma	2	6	6	6	6	6	
Phytophthora multivora	2	6	6	6	0	0	
Phytophthora palmivora	1	3	3	3	3	3	
Phytophthora plurivora	1	3	3	3	3	3	
Phytophthora polonica	1	3	3	3	3	3	
Phytophthora pseudosyringae	1	3	3	3	3	3	
Phytophthora ramorum	2	6	6	6	6	6	
Phytophthora rosacearum	2	6	6	6	0	3	
Phytophthora rubi	1	3	3	3	3	3	
Phytophthora syringae	1	3	3	3	3	3	
Phytophthora taxon Raspberry	1	3	3	3	3	3	
Phytophthora taxon Walnut	2	6	6	6	6	6	
Pythium folliculosum	1	3	3	3	3	3	
Total	43	129	128	129	101	117	
			(99.2%)	(100%)	(78.3%)	(90.7%)	
Notes.

a Secure genus identification, probable species identification.

b Probable genus identification.

From genus to strain using the custom library

In total, 42 samples within the genus Phytophthora and 1 sample belonging to the related genus Pythium, which was included in the collection for quality control purposes, were compared. The MSP spectra are compared in the form of log score values in Table S2. All 42 samples matched their own MSPs (biological repeats) measured on another day with log scores greater than 2.3. For one sample (P. gonapodyides 419.10), the highest match category had a score of 2.3 with only one biological replicate and a second score of 2.2, which is satisfactory for probable species identification. The results of the reliability of identification of all species and strains are summarized in Table 2. Two limit values were determined according to the manufacturer’s recommendations. At a limit value of 2, the confidence of strain identification was 78.3%; at a lower limit value of 1.7, the success rate was 90.7%.

Comparison within a species

The spectra of different strains of P. lacustris analyzed in flexAnalysis (Fig. 2) showed protein ion peaks in the 6,500 and 14,500 m/z regions for all samples, but the intensities differed among the samples. All spectra were smoothed and baseline aligned prior to evaluation. For P. lacustris, reliable identification of strains from different locations and hosts was also achieved within the species using the log score as shown in Table 3. The MSP dendrogram based on comparison of the MALDI-TOF mass spectra is shown in Fig. 3.

Figure 2 Protein spectra of different strains of P. lacustris.

Table 3 Identification of P. lacustris from different locations and hosts.

Strain	Database match*	
P. lacustris 361.09	361.09/2	385.10/2	295.09/1	385.10	295.09/2	
score	2.796	2.668	2.593	2.589	2.464	
P. lacustris 295.09	295.09/2	361.09/2	361.09/1	295.09	361.09	
score	2.759	2.64	2.591	2.536	2.362	
P. lacustris 385.10	385.10/2	385.10	361.09/1	385.10/3	361.09/2	
score	2.749	2.599	2.586	2.578	2.445	
Notes.

* The number after the slash indicates the number of biological replicates.

Figure 3 MSP dendrogram of P. lacustris from different locations.

The MSP spectra of P. alni subsp. uniformis 239.08 were very similar to those of P. cambivora, P. alni subsp. alni and P. rubi. The log scores ranged from 2.4 to 2, indicating some association among these species. However, the dendrogram placed these species in different clusters (Fig. 4). There was also a match of log scores at the species level between P. cactorum 862.17 and P. hedraiandra. For the genus Pythium, only consistency among biological replicates was observed. The log score compared to Phytophthora was less than 1.6 (Table 4)

Figure 4 Distribution of different species with similar spectral profiles according to MSP dendrogram cluster analysis.

Table 4 Identification of Pythium with Phytophthora library.

Strain	Database match*	
Pythium folliculosum890.17	Pythium folliculosum 890.17/2	Pythium folliculosum 890.17/3	P. rubi 734.14/1	P. alni subsp. alni 240.08	P. cambivora 286.09/2	
Score	2.59	2.548	1.578	1.478	1.468	

Discussion

The genera Phytophthora and Pythium include many economically important species that have been placed in Kingdom Chromista (H Ho, 2018). Many Phytophthora species are relatively easy to identify, but overlapping morphological features and intraspecific variability can make genus identification difficult (Martin et al., 2012) in the absence of genetic methods. Cooke et al. (2000) developed the first molecular phylogeny for the genus Phytophthora by analyzing sequences of the ITS region. Several multilocus phylogenies were subsequently constructed that divided Phytophthora species into 10 phylogenetically well-supported clades and several subclades (Kroon et al., 2004; Blair et al., 2008; Martin, Blair & Coffey, 2014; Yang, Tyler & Hong, 2017). The clade affiliations of the species used in this study are given in Table S1. After genetic validation using COI and ITS, isolates of these species were grown in liquid medium and subsequently analysed by MALDI-TOF MS to create a database. The MSP database (http://doi.org/10.5281/zenodo.4271724) created in this study can be used as a complement to existing datasets for fast and reliable identification of Phytophthora.

MALDI-TOF MS is a fast and direct method of analysis that requires no chromatographic separation (Galeano Garcia et al., 2018) and is much less expensive than DNA sequencing (Del Castillo-Múnera et al., 2013). Recent research has demonstrated the usefulness of MALDI-TOF MS for the rapid characterization of clinical pathogens, lactic acid bacteria, nonfermenting bacteria, fungi, plant-parasitic nematodes, and environmental bacteria (Ahmad, Babalola & Tak, 2012). Moreover, studies using a wide range of microbial species suggest that MALDI-TOF MS has better diagnostic accuracy than conventional biochemical techniques (Ahmad, Babalola & Tak, 2012). MALDI-TOF MS reference spectra and analytical tools are usually proprietary, in contrast to the wide range of public repositories of openly accessible or downloadable data and tools for DNA-based identification. For example, in 2012, Bruker Daltonics launched Filamentous Fungi Library 1.0, a commercial MS library for fungal identification (Schulthess et al., 2014). The production and integration of in-house MS databases for open access would facilitate the creation of a truly open platform for MALDI-TOF MS-based species identification (Drissner & Freimoser, 2017) and promote the use of MALDI-TOF MS biotyping in other areas of diagnostics, such as plant pathogens or food-important bacteria, fungi and yeasts (Drissner & Freimoser, 2017).

Most studies using MALDI-TOF MS for microbial identification refer to the Bruker system, which is proprietary but the most widespread system (Normand et al., 2017). Studies of the identification of filamentous fungal species using the Bruker system have focused on the genera Fusarium, Aspergillus, Penicillium and Ramularia, which are potentially dangerous to humans (Boekhout et al., 2015). However, analyses of dermatophyte genera using the Saramis, Vitek MS or Andromas systems have also been published. In human clinical microbiology, MALDI-TOF MS has primarily been used for the identification of bacteria and yeasts using in-house MS databases of human pathogens (Gräser, 2014) Lasch, Stämmler & Schneider, 2018; Paul et al., 2019; Papalia et al., 2020) and environmental isolates of Burkholderia and related genera (Fergusson et al., 2020).

Although MALDI-TOF MS is becoming increasingly popular as a tool for clinical diagnosis, the lack of data on non-clinical microorganisms has limited its use in microbial ecology (Rahi, Prakash & Shouche, 2016). So far very little attention has been paid to plant pathogens such as Phytophthora. Even organisms commonly found in environmental, agricultural, or food samples are often not recognized by standard, commercial MALDI-TOF MS systems (Drissner & Freimoser, 2017). The available MBT Library BDAL 9.0 (Bruker Daltonics, DE), which includes 8,468 MSPs, and the MBT Filamentous Fungi Library V1.0 (365 MSPs) do not contain any representatives of the genus Phytophthora. In a previous study, an in-house MS database was used to accurately identify Pythium insidiosum isolates and properly differentiate them from other filamentous fungi in patient samples (Mani et al., 2019), thus demonstrating that MALDI-TOF MS can be used for the accurate and rapid culture identification of Pythium insidiosum in the clinical laboratory. Our dataset contained Pythium folliculosum as a control strain that can be reliably distinguished from other Phytophthora species. As a contrast to this technical approach, ongoing studies suggesting that dogs can also rapidly and reliably distinguish plants infected with Phytophthora or Pythium species (Swiecki et al., 2018; Oliver et al., 2020).

The MSP spectra of P. hedraiandra and P. cactorum were similar, consistent with phylogenetic studies that have indicated that P. hedraiandra is closely related to P. cactorum or basal to the cluster of P. cactorum and P. pseudotsugae (Blair et al., 2008; Martin, Blair & Coffey, 2014; Yang, Tyler & Hong, 2017); all of these species belong to Subclade 1a. Similarity of the MSP spectra was also observed for P. alni subsp. alni, P. alni subsp. uniformis, P. cambivora and P. rubi, which are phylogenetically closely related and belong to Subclade 7a (Yang, Tyler & Hong, 2017). Despite these similarities, we were able to discriminate 43 genetically and phenotypically distinct Phytophthora strains by native Bruker Biotyper methods.

Some in house databases based on profile mass spectra was developed for identification of species of Xanthomonas (Sindt et al., 2018), Rhizopus (Boekhout et al., 2015), Tuber (El Karkouri et al., 2019). MALDI-TOF MS has also been used for the in situ identification of plant-invasive bacteria, including rhizobia (Ziegler et al., 2012) and Acidovorax citrulli isolates (Bergsma-Vlami, 2018) and late blight in asymptomatic tomato plants (Galeano Garcia et al., 2018).

MALDI-TOF MS-based identification of P. insidiosum revealed differences in protein spectra between geographically different isolates (Mani et al., 2019). While our samples were also taken from geographically different habitats and hosts, the results clearly showed that these differences did not affect the reliability of identification. Notably, we were able to use our constructed database to distinguish P. palmivora and P. infestans with high reliability. Previous protein profiling of these two species by two-dimensional gel electrophoresis showed that the overall proteome maps were similar; equivalent numbers of spots were detected for the two species, and 30% of the protein spots had similar or identical positions in the 2D gels (Shepherd, Van West & Gow, 2003). Thus, our results suggest that the use of MALDI-TOF MS biotyping combined with simple sample processing and subsequent comparison of spectra with a database is effective for the accurate and rapid identification of Phytophthora genera and species. In addition, due to its simplicity, this method could be useful for laboratories with limited expertise in mycology (Mani et al., 2019).

Conclusions

This study demonstrated that MALDI-TOF MS can be used to identify phytopathogens of the genus Phytophthora. None of the Phytophthora isolates have previously been included in a commercial library. Our updated library allowed the correct identification of 117/129 (90.6%) Phytophthora isolates at the species level and 100% of isolates at the genus level. The success rate within species was 78.3% at a cut-off value of 2 and 90.7% at the lower limit of 1.7. In addition, protein extraction using ethanol, formic acid and acetonitrile was quick and effective. Further development of MALDI-TOF MS-based identification of fibrous oomycetes will require continuous updating of current commercial databases and devices as well as the construction of new databases for specialized research purposes. Another prospect for development is the potential detection of typical MS markers of Phytophthora species directly in infected plant material.

Supplemental Information

Supplemental Information 1 Flow diagram displaying the route for identification from isolation from plants into MALDI analyses

Click here for additional data file.

Supplemental Information 2 List of isolates and GenBank identification

Click here for additional data file.

Supplemental Information 3 Full comparison

Click here for additional data file.

The authors wish to thank Dawn Schmidt for language editing.

Additional Information and Declarations

Competing Interests

Author Contributions

Data Availability

The authors declare there are no competing interests.

Matěj Božik conceived and designed the experiments, analyzed the data, prepared figures and/or tables, authored or reviewed drafts of the paper, and approved the final draft.

Marcela Mrázková, Karolína Novotná and Markéta Hrabětová performed the experiments, analyzed the data, prepared figures and/or tables, authored or reviewed drafts of the paper, and approved the final draft.

Petr Maršik and Pavel Klouček conceived and designed the experiments, authored or reviewed drafts of the paper, and approved the final draft.

Karel Černý conceived and designed the experiments, prepared figures and/or tables, authored or reviewed drafts of the paper, and approved the final draft.

The following information was supplied regarding data availability:

All sequences obtained in this study are available at NCBI GenBank: MT293575, MT293576, MT293577, MT350310, MT350311, MT350312, MT350313, MT350314, MT350315, MT350316, MT350317, MT350318, MT350319, MT350320, MT350321, MT293578, MT350322, MT350323, MT293579, MT350324 MT293580, MT350325, MT293581, MT350326, KJ567086, KJ567083, MT350327, MT350328, KJ567085, KJ567082, MT350329, MT350330, MT350331, MT293582, MT293583, MT350332, MT350333, MT350334, KX378014, KX378013, MT350335, MT293584, MT293585, MT350336, MT293586, MT350337, MT293587, MT350338, MT293588, MT293589, MT293590, MT350339, MT293591.

The MALDI-TOF MS reference library and associated data are available at Zenodo: Božik Matej, Mrázková Marcela, Novotná Karolína, Hrabětová Markéta, Maršík Petr, Klouček Pavel, & Černý Karel (2020). Custom main spectra (MSP) library for the identification of Phytophthora species by MALDI-TOF MS (Version 1) [Data set]. Zenodo. http://doi.org/10.5281/zenodo.4271724.

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
