# Peer review of "MALDI-TOF MS as a method for rapid identification of Phytophthora de Bary, 1876"

_PeerJ, doi:10.7717/peerj.11662_

## Round 0.1 · original submission · Minor Revisions

The reviewers suggested minor revisions for your article. Please, address their comments and resubmit your manuscript within the the next 21 days.

Reviewer 1 ·

Basic reporting

The study by Bozik and colleagues shed light into the application of routine MALDI-TOF mass spectrometry for the rapid identification of Phytophthora species.

The manuscript is clear, straightforward and the data support the primary hypothesis.

Experimental design

The experiments have been conducted diligently with appropriate statistical analyses.

Validity of the findings

The data support the research hypothesis.

Additional comments

It would be of importance to include a flow diagram displaying the route for identification from isolation from plants into MALDI analyses.

Reviewer 2 ·

Basic reporting

Title
1.MALDI-TOF MS as a method for rapid identification of
Phytophthora species
ex. Please inform author and year. ex. Phytophora Bary, 1986

2. Correct throughout the text (M / Z), correct is italic.

Experimental design

MALDI-TOF MS as a method for rapid identification of Phytophthora species.

It is a tool that is evolving mainly in the microbiological part, fungi and Bacteria. The article is to be congratulated both on the materials and methods. They used a (Byotiper) database to compare with spectra, which is essential for verification.

MALDI TOF tool has an advantage over molecular biology, as it is very fast and efficient, that's why I find the article very innovative and attractive.


Material and method well based, correctly describing the steps of the protocol process.

Validity of the findings

The conclusions are well formulated, linked to the original research question and limited to supporting results.

---

## Round 0.2 · accepted · Accept

The authors have incorporated the suggested changes provided by the reviewers.